# Ball-Milling Preparation of the Drug–Drug Solid Form of Pioglitazone-Rosuvastatin at Different Molar Ratios: Characterization and Intrinsic Dissolution Rates Evaluation

**DOI:** 10.3390/pharmaceutics15020630

**Published:** 2023-02-13

**Authors:** M. Fernanda Muñoz Tecocoatzi, José C. Páez-Franco, Kenneth Rubio-Carrasco, Alejandra Núñez-Pineda, Alejandro Dorazco-González, Inés Fuentes-Noriega, Alfredo R. Vilchis-Néstor, Lilian I. Olvera, David Morales-Morales, Juan Manuel Germán-Acacio

**Affiliations:** 1Red de Apoyo a la Investigación, Coordinación de la Investigación Científica-UNAM, Instituto Nacional de Ciencias Médicas y Nutrición SZ, Ciudad de Mexico C.P. 14000, Mexico; 2Laboratorio de Biofarmacia, Departamento de Farmacia, Facultad de Química, UNAM, Ciudad de Mexico C.P. 04510, Mexico; 3Centro Conjunto de Investigación en Química Sustentable CCIQS UAEM-UNAM, Carretera Toluca-Atlacomulco km 14.5, Toluca C.P. 50200, Mexico; 4Instituto de Química, Universidad Nacional Autónoma de Mexico, Circuito Exterior, Ciudad Universitaria, Ciudad de Mexico C.P. 04510, Mexico; 5Instituto de Investigacioes en Materiales, Universidad Nacional Autónoma de Mexico, CU Coyoacán, Ciudad de Mexico C.P. 04510, Mexico

**Keywords:** drug–drug coamorphous, mechanochemical reactions, intrinsic dissolution experiments, rosuvastatin, pioglitazone hydrochloride

## Abstract

Ball-milling using neat grinding (NG) or liquid-assisted grinding (LAG) by varying the polarity of the solvents allowed access to various drug–drug solid forms of pioglitazone hydrochloride (PGZ·HCl) and rosuvastatin calcium (RSV). Using NG, the coamorphous form was formed from the reaction of pioglitazone hydrochloride (PGZ·HCl) and rosuvastatin calcium (RSV) in a 2:1 molar ratio. The formation of the expected coamorphous salt could not be corroborated by FT-IR, but DSC data showed that it was indeed a single-phase amorphous mixture. By varying the molar ratios of the reactants, either keeping PGZ·HCl constant and varying RSV or vice versa, another coamorphous form was obtained when a 1:1 molar ratio was employed. In the case of the other outcomes, it was observed that they were a mixture of solid forms coexisting simultaneously with the coamorphous forms (1:1 or 2:1) together with the drug that was in excess. When RSV was in excess, it was in an amorphous form. In the case of PGZ·HCl, it was found in a semicrystalline form. The intrinsic dissolution rates (IDRs) of the solid forms of PGZ·HCl-RSV in stoichiometric ratios (1:1, 2:1, 1:4, 6:1, and 1:10) were evaluated. Interestingly, a synchronized release of both drugs in the dissolution medium was observed. In the case of the release of RSV, there were no improvements in the dissolution profiles, because the acidic media caused the formation of degradation products, limiting any probable modification in the dissolution processes. However, the coamorphous 2:1 form exhibited an improvement of 1.03 times with respect to pure PGZ·HCl. It is proposed that the modification of the dissolution process of the coamorphous 2:1 form was limited by changes in the pH of the media as RSV consumes protons from the media due to degradation processes.

## 1. Introduction

Statins are the most recurrent pharmaceutical agents used for lipid disorders [1]. These drugs inhibit the production of HMG-CoA, an essential metabolite in cholesterol biosynthesis [1]. Rosuvastatin calcium (RSV) (Figure 1) is a drug from the statin family with a wide variety of crystalline states and can also exist in an amorphous form [2]. In addition, RSV is a BCS Class II drug, so it exhibits a low aqueous solubility of 0.01 mg/mL and low bioavailability [3]. As about 40% of the medicines offered on the market have solubility problems (BSC Class II), the scientific community has taken this as a challenge to overcome [4]. The amorphization of a drug is the easiest way to increase its solubility [5]. It is well known that an amorphous state typically possesses properties of higher solubility and a higher rate of dissolution than its crystalline counterpart [6]. Crestor^®^ (the commercial form of the RSV) is distributed on the market in an amorphous form [2]. One of the drawbacks of the amorphization of a drug is the metastable phases that possess high internal energy and enhanced molecular mobility and that tend to recrystallize [7]. This shortens the shelf life of the drug. Given this, other approaches have been designed to improve the aqueous solubility of drugs [8]. Among them, liposomes, nanoparticles, self-emulsifier drug delivery systems (SMEDDSs), hydrotrophy, cyclodextrin complexation, cosolvency, chemical modification, solid lipid nanoparticles, and solid dispersions (SDs) stand out [4]. According to a recent review, SDs are considered the “dispersion of one or more active substances in an inert carrier prepared by melting, dissolution or melting-dissolution” [9]. SDs can generally be classified according to their nature into eutectic mixtures, solid solutions, glass solutions, and glass suspensions [9].

In this regard, the preparation of polymer-based amorphous solid dispersions (PASDs) is recurrently used to avoid the tendency of a drug to recrystallize [10,11]. Thus, adding a polymer (hydrophilic) entraps the drug (hydrophobic) in its matrix, forming the PASD. The polymers used present high glass transition temperature (T_g_) values, which cause the increase in the T_g_ of the amorphous drug. The polymers function as stabilizers that decrease the internal molecular mobility, inhibiting nucleation and crystal growth and slowing down the recrystallization process. However, PASDs have the following disadvantages: (1) large amounts of polymer are required to ensure molecular mixing with the drug, leading to oversized dosage units; (2) PASDs are susceptible to moisture and heat, causing its molecular destabilization; (3) they can sometimes present manufacturing process problems as they exhibit a sticky nature [9]. The preparation of coamorphous systems can be an alternative solution to these problems. This approach (coamorphous, which belongs to the family of SDs) is based on combining two or more low-molecular-weight components to form a homogeneous amorphous single phase [9]. It is emphasized that the components are in close stoichiometric relationships, contrary to what is seen in PASDs. In the case of pharmaceutical coamorphous forms, incorporating a coformer or another drug allows the increase in T_g_, limiting molecular mobility and avoiding recrystallization [9]. Additionally, the coamorphous systems present a great variety of non-covalent bonds (hydrogen bond, π–π, ionic interactions) that allow them to have great molecular stability that prevents recrystallization. However, some coamorphous drug–drug systems (e.g., simvastatin/glipizide) have been reported in which no intermolecular interactions between the components were detected [12]. Glipizide acted as an antiplasticizing agent. In the case of PASDs, the variety of intermolecular drug–polymer interactions is more limited, which probably causes the recrystallization process to be faster.

Based on all the above, in this paper, we present the preparation of the drug–drug coamorphous PGZ·HCl-RSV (PGZ·HCl: pioglitazone hydrochloride, Figure 1) prepared by mechanochemical methods (ball-milling) [8], varying the stoichiometric proportions. We seek to explore various molar ratios between the two drugs as it has been mentioned that the formation of the coamorphous form in a 1:1, 2:1, or 1:2 ratio does not guarantee that it is the therapeutically relevant dose [13]. Reactions are performed using neat grinding (NG) or liquid-assisted grinding (LAG) [8]. PGZ·HCl is used as the other coformer drug because it has been mentioned that thiazolidinediones drugs combined with statins may have a beneficial synergistic effect in the treatment of diabetic patients with dyslipidemia [14,15]. In this way, we want to evaluate the solubility behavior and dissolution properties of these solid forms. It is highlighted that, to our best knowledge, this would be the first example of preparing a coamorphous system containing RSV. Within the SD family, the PASD using Eudragit^®^ (hydrophilic polymer) loaded with RSV has recently been published. These compositions showed changes in the solubility of RSV [4].

## 2. Materials and Methods

### 2.1. Materials

All the pharmaceutical reagents were purchased from Tokyo Chemical Industry™ (Portland, OR, USA) (PGZ·HCl: P1901, >98%) or Merck-Supelco Mexico™ (Naucalpan de Juárez, Mexico) (RSV: PHR1928, Pharmaceutical Secondary Standard, certified reference material) and were used as received. The solvents were purchased from Tecsiquim™ (Toluca de Lerdo, Mexico) and were used as received. The methanol used in the mobile phase for the determination of the dissolution profiles was HPLC grade. Other solvents used were reagent-grade.

### 2.2. Methods

#### 2.2.1. NG or LAG Solvent-Screening (Stoichiometry Ratio 2:1)

NG or LAG solvent-screening for the preparation of the solid forms was performed using a Planetary Micro Mill Pulverisette^TM^ 7 Fritsch device (Idar-Oberstein, Germany) [8,16]. PGZ·HCl (237.00 mg, 0.6032 mmol) and RSV (150.00 mg, 0.1498 mmol) were ball-milled in a stoichiometric ratio of 2:1. It should be remembered that for the number of moles used in the reaction, two RSV^-^ molecules are released. For every LAG experiment, 100 μL of solvent was added. The solvents used were hexane, ethyl acetate (AcOEt), ethanol (EtOH), and water. Stainless-steel bowls of 20 mL containing 10 stainless-steel balls (10 mm diameter) were used. The NG or LAG experiments were carried out at 600 rpm for 30 min.

#### 2.2.2. Evaluation of the Formation of the Multicomponent Salt PGZ-RSV (EtOH, Stoichiometric Ratio 2:1) at Different Grinding Times

Ball-milling studies of PGZ-RSV (LAG with EtOH) were carried out by lengthening the milling times and applying heat treating (H.T.) at 139 °C. PGZ·HCl (237.00 mg, 0.6032 mmol) and RSV (150.0 mg, 0.1498 mmol) were ball-milled. Stainless-steel bowls of 20 mL containing 10 stainless-steel balls (10 mm diameter) were used. The experiments were carried out at 600 rpm. A sample was periodically withdrawn to be analyzed by XRPD at 30 min, 60 min, 90 min, and 120 min. To the sample at 120 min, H.T. was applied. H.T. was carried out using an OMH60 Heratherm Thermo Scientific^®^ (Santa Clara, CA, USA) mechanical convection oven. An initial temperature of 50 °C was started, with a heating rate of 10 °C/min, and the maximum temperature reached of 139 °C was held for 1 h. Once the powders were removed from the oven, they were ground while hot in the mortar for 30 min (2 h + H.T.). 

#### 2.2.3. Evaluation of the Amorphization Ability of the PGZ·HCl

Starting with 250 mg of PGZ·HCl, it was ball-milled under NG. It should be noted that approximately 40 mg of sample was periodically withdrawn (30, 60, 90, 120, and 150 min) to be characterized by XRPD and DSC-TGA. Stainless-steel bowls of 20 mL containing 10 stainless-steel balls (10 mm diameter) were used. The experiments were carried out at 600 rpm.

#### 2.2.4. Evaluation of the Formation of the PGZ·HCl-RSV Solid Forms (1:1, 1:2, 1:4, 1:6, 1:8, and 1:10)

Table 3 shows the amounts in mg of each drug used to prepare the corresponding solid forms. For each solid form, NG was used for 30 min at 600 rpm. Stainless-steel bowls of 20 mL containing 10 stainless-steel balls (10 mm diameter) were used. In the case of the solid form (1:1), the ball-milling times were extended up to 150 min. Samples were periodically withdrawn to be able to characterize them (XRPD and DSC-TGA) at different milling times (30, 60, 90, 120, and 150 min).

#### 2.2.5. Evaluation of the Formation of the PGZ·HCl-RSV Solid Forms (4:1, 6:1, 8:1, and 10:1)

Table 5 shows the amounts in mg of each drug used to prepare the corresponding solid forms. For each solid form, NG was used for 30 min at 600 rpm. Stainless-steel bowls of 20 mL containing 10 stainless-steel balls (10 mm diameter) were used.

#### 2.2.6. Evaluation of the Amorphization of the RSV

Starting with 250 mg of RSV, it was ball-milled under NG. It should be noted that approximately 40 mg of sample was periodically withdrawn (30, 60, 90, 120, and 150 min) to be characterized by DSC-TGA. Stainless-steel bowls of 20 mL containing 10 stainless-steel balls (10 mm diameter) were used. The experiments were carried out at 600 rpm.

#### 2.2.7. Thermal Analysis

Two different types of equipment were used interchangeably to carry out the DSC and TGA experiments. A simultaneous thermal analyzer Netzsch STA 449 F3 Jupiter was used. A DSC Q100 V9.9 Build 303 (TA instruments) was used. In addition, TGA Q5000 V3.17 Build 265 (TA instruments) equipment was employed. The samples were placed (2–4 mg) in sealed non-hermetic aluminum pans and were scanned at a heating rate 20 °C/min from 30 to 400 °C under a dry nitrogen atmosphere. The calculated T_g_ values of the synthesized coamorphous form were predicted by employing the Fox equation [17]
$$\frac{1}{T_{\mathrm{gmix}}}=\frac{w_{1}}{T_{g1}}+\frac{w_{2}}{T_{g2}}$$

T_g1_ and T_g2_ are glass transition temperatures of the components (RSV: 72.8 °C) and (PGZ·HCl: 64.4 °C), respectively [18]; w_1_ and w_2_ are the corresponding weight fractions of the components; T_gmix_ is the glass transition of the coamorphous mixture. The RSV T_g_ value was obtained from our results (Appendix A). 

#### 2.2.8. XRPD

XRPD experiments were carried out in a Bruker D8 Advance diffractometer with Bragg–Bretano geometry, Cu Kα radiation (1.54060 Å), and a Linxeye detector. Each sample was measured by a continuous scan between 5 and 60° in 2θ, with a step time of 151.19°/min and step size of 0.0198°. 

#### 2.2.9. Scanning Electron Microscopy Studies (SEM) 

The morphology of each solid form was evaluated by SEM on a JEOL (JSM-6610) microscope. For sample preparation, the specimen was dried and fixed on a stub with carbon double-stick tape and then coated with gold for 90 s under vacuum using a Denton IV sputtering chamber.

#### 2.2.10. Intrinsic Dissolution Studies

The intrinsic dissolution constants (K_int_) were determined according to the conditions established in the United States Pharmacopea (USP) [19]. The experiments were carried out using tablets, prepared with a hydraulic press with a pressing force of 220 kg/cm^2^. Dissolution rates were determined using Wood’s apparatus according to the USP. Dissolution profiles were performed using hydrochloric acid and potassium chloride buffer (pH = 2.0) as the established dissolution medium for PGZ·HCl, USP [19]. The experiments were carried out in triplicate at 37 °C under constant stirring (100 rpm) in a constant volume of 900 mL. The profiles were quantified using an Agilent 1260 series Infinity II HPLC equipment, with a high-performance autosampler (G1367E) under the following chromatographic conditions: mobile phase MeOH, H_2_O, and 0.01 M H_3_PO_4_, pH = 2 (60:25:15), with a flow of 1.500 mL/min, using a Zorbax Eclipse XDB-C18 column, 4.6 × 150 mm, with a particle size of 5 μm, and with a diode array UV-vis detector, and samples were measured at a wavelength of 238 nm.

#### 2.2.11. Saturation Solubility Experiments

An excess amount of powder (RSV or any solid form) was weighed to approximately 20.0 mg and dissolved in a fixed-volume vial in the USP recommended medium for PGZ HCl (10.0 mL) to obtain a concentration of approximately 2 mg/mL [19]. The vial was shaken magnetically for 24 h at 20 °C. Aliquots were taken at 0 h and subsequently passed for 24 h through 0.45 µm filters, adequately diluted, and quantified by HPLC (Agilent 1260 infinity II) using a calibration curve. The experiments were performed in triplicate.

## 3. Results

### 3.1. NG and LAG Solvent-Screening (Stoichiometry 2:1)

The molar ratio of (2:1) of PGZ·HCl-RSV is used as a model reaction in the NG and LAG screening-solvent (Table 1). It is noteworthy that this stoichiometry is used to observe if the coamorphous salt is formed. LAG is used to explore the influence of the solvents (varying the polarity) to see what effect it has on the formation of the solid form. The binary solid forms obtained are analyzed by XRPD (Figure 1). In the first instance, a high crystallinity is not observed, as an amorphous contribution is observed in all outcomes. It is also observed that there are not many differences in obtaining the solid form either by LAG (varying the solvents). In the case of the NG and AcOEt outcomes, they are the ones with a profile most like a coamorphous formation. The presence of CaCl_2_ is not observed in any of the solid forms, an expected byproduct in case the multicomponent salt is formed. From these results, we can affirm that it is impossible to form the multicomponent salt with high crystallinity. Trying to obtain the multicomponent salt, we extend the reaction times and apply H.T. to the sample LAG in EtOH. This outcome shows a higher crystallinity compared to the others. The results obtained are analyzed by means of XRPD (Appendix A). It is immediately observed that from the first 30 min of ball-milling, the solid phase is formed and that, even extending the reaction times to 2 h and applying H.T., the solid phase does not undergo any change as it is observed that the diffraction pattern persists in each case. It should be emphasized that in these studies extending the milling times, the presence of CaCl_2_ is not observed.

These solid forms are also analyzed by DSC (Figure 2 and Table 1). The individual thermograms of all these results, DSC-TGA, and the pure drugs are shown in (Appendix A).

Observing the thermograms obtained from the preparation by NG or LAG solvent screening, they are different from the profiles of the pure components. At first glance, NG presents three different thermal events, presenting a glass transition temperature (T_g_: 52.55 °C), crystallization temperature (T_c_: 116.26 °C, exo), and melting temperature (T_m_: 164.27 °C). It should be noted that this T_g_ event presents an enthalpy relaxation endotherm. A classic T_g_ signal does not exhibit this small endotherm. The appearance of this event implies an enthalpic relaxation (ΔH) due to the aging or relaxation of the amorphous sample. This endothermic enthalpy relaxation effect will increase as the solid form ages or relaxes over time [20]. The presence of T_c_ in the DSC is attributed to PGZ·HCl. As is discussed in Section 3.1.1, an attempt is made to amorphize the PGZ·HCl by ball-milling, but this is unsuccessful. The extent of relaxation that one of the components can reach is dependent on the enthalpy change, and this is regarded as an equilibrium of going from a glassy state to a supercooled liquid [21]. Thus, the contribution of endothermic enthalpy relaxation is significant (5.21 J/g). Therefore, the rate at which PGZ·HCl approaches a relaxed state occurs immediately after the glass transition is overcome. RSV cannot prevent recrystallization, as it is not capable of keeping PGZ·HCl in an amorphous state. The T_g_ value of the pure PGZ·HCl cannot be obtained from the first derivative of the DSC; otherwise, with respect to pure RSV (Appendix A), this indicates that PGZ·HCl by itself does not have the ability to amorphize, at least by ball-milling. It has been reported that sucrose crystallization can be inhibited by adding additives (poly(vinylpyrrolidone), poly(vinylpyrrolidone-co-vinyl-acetate), dextran trehalose, etc.) as the polymers establish intermolecular interactions with the carbohydrate. At the molecular level, the additives impart subtle differences in molecular mobility in the blend that are not reflected in the T_g_ value [22]. In this case, Zografi et al. mention that molecular mobility should not be seen as the only factor controlling the inhibition of sucrose crystallization. They indicate that thermodynamic and geometric factors that control nucleation must be considered [22]. This recrystallization process of one of the components within coamorphous systems has already been seen previously for tadalafil-repaglinide [23].

Observing a single T_g_ value indicates the miscibility of the components forming an amorphous single-phase (coamorphous), as the presence of two T_g_ values denotes that the constituents would be separated into two phases [24]. Additionally, the diffractogram profile of the solid form obtained by NG shows a large amorphous contribution, indicating the formation of the coamorphous 2:1 form (Figure 1).

The value of T_g_ is calculated (67.41 °C) with the Fox equation (Section 2.2.7). It has been mentioned that when the T_gmix_ prediction equations deviate from the experimental value, it is because they do not contemplate the intermolecular interactions between the components of the mixture [25,26]. Positive deviations above the predicted values of the Fox equation indicate that the two components interact at the molecular level [26,27,28]. In our case, the opposite occurs; the calculated T_g_ value is above the experimental value (52.55 °C). In fact, the calculated value is very close to the reported PGZ·HCl T_g_ value (64.4 °C). In this way, it is suggested that apart from the fact that the Fox equation does not contemplate the interactions between the components, we believe, as Zografi indicates, other factors must contribute that cause this type of deviation between the calculated and experimental values [22]. The contribution of enthalpy relaxation is not contemplated in this equation, which must indicate that the prediction does not correlate well.

In the case of the solid form prepared by LAG in AcOEt, it presents the values of T_c_: 110.11 °C and T_m_: 164.98 °C, but the presence of T_g_ is missing. The lack of T_g_ may indicate that under these preparation conditions, the components does not present miscibility to form a coamorphous form [10,11]. In the case of the other solid forms (hexane, EtOH, and water), they do not present T_g_ or T_c_, only T_m_ (Table 1). In the case of the solid forms prepared by LAG (hexane and EtOH), they present endothermic events at 72.8 and 63.1 °C, respectively, which are attributed to water molecules of hydration according to TGA. The corresponding thermogram for LAG (water) presents two endotherms that indicate the formation of a physical mixture as, under these conditions, a single-phase solid form is not formed, due to the presence of two separated endotherms. This indicates that ball-milling with NG favors the formation of the coamorphous form. In this way, we can observe the potential that ball-milling can have to access to a diversity of solid forms. In this case, in this system, exploring it by NG or LAG solvent-screening, the following is obtained: NG (coamorphous), water (physical mixture), and EtOH and hexane (not defined). It should be emphasized that in the case of the solid form, AcOEt does not completely favor the formation of the coamorphous form as is achieved in NG. Thus, ball-milling can represent a great tool in the ex situ study for the proposal of reaction pathways and the reaction mechanisms [29]. 

Participating interactions in these solid forms are assessed using FT-IR. Initially, the formation of the synthon -COO^-^···^+^H-N_pyr_ is expected, which is a charge-assisted hydrogen bond (Figure 3). 

Spectra are analyzed in the interval of 2000–1300 cm^−1^ (Figure 4). The full spectra of all outcomes are found in Appendix A. The vibrational modes −C=O_PGZ+_ (1744 and 1690 cm^−1^) [30] and −C=O_RSV−_ (1543 cm^−1^) [31] of the pure drugs are evaluated in Table 2. From this table, calculating Δν of any of the vibration modes of all the samples indicates that there are no changes in the participating intermolecular interactions, except for −C=O_PGZ+_ (b), which presents a Δν of 11 cm^−1^. Not observing any shift comparing the vibration mode −C=O_RSV−_ of pure RSV and any of the solid forms obtained indicates that the synthon −COO^−^···^+^H-N_pyr_ is not established. This evidence confirms that the solid form obtained by NG is not a coamorphous salt. Therefore, it can be suggested that a coamorphous system is formed, where the two components present miscibility forming an amorphous single-phase.

As mentioned above, a coamorphous system (simvastatin/glipizide) has already been reported where there is reported to be no indication of intermolecular interactions between the components [12].

From this, NG is used to prepare the solid forms of PGZ·HCl-RSV trying to form a coamorphous form in different stoichiometric ratios. 

#### 3.1.1. Evaluation of the Formation of the PGZ·HCl-RSV Solid Forms (1:1, 1:2, 1:4, 1:6, 1:8, and 1:10)

We first evaluate this set of binary solid forms to prove if a coamorphous system is formed. In this case, the stoichiometric ratio of PGZ·HCl remains constant, and RSV varies. Initially, the different proportions are evaluated by means of XRPD (Figure 5).

It is observed that in any of these molar ratios, an intense halo is observed at 2θ ~20°. This indicates at first glance that the coamorphous system is formed for any case. Only in the case of the stoichiometry 1:1 is the presence of reflections corresponding to PGZ·HCl incipiently observed in the halo. In this sense, we extend the milling times (30, 60, 90, 120, and 150 min) to prove if this favors the complete formation of the coamorphous form and if the reflections corresponding to PGZ·HCl disappear. DSC-TGA and XRPD results are seen in Appendix A. Individual thermograms of DSC-TGA of the solid-phase PGZ·HCl-RSV 1:1 at different milling times are presented in Appendix A. Extending the ball-milling times helps the reflections of PGZ·HCl to disappear after 1 h of milling (Appendix A). Regarding the DSC (Appendix A), it is observed that within the first 30 to 90 min of milling, the appearances of T_g_ (30 min: 55.41 °C; 60 min: 57.77 °C; 90 min: 57.27 °C), T_c_ (30 min: 111.46 °C; 60 min: 120.72 °C; 90 min: 128.52 °C), and T_m_ (30 min: 156.39 °C; 60 min: 155.39 °C; 90 min: 155.49 °C) are observed. 

Again, it is proposed that the appearance of T_c_ in the DSCs (30, 60, and 90 min) is due to the presence of semi-crystalline PGZ·HCl in this coamorphous mixture. This is because PGZ·HCl is reluctant to amorphize by ball-milling actions. The XRPD and DSC-TGA results of these amorphization experiments are shown in Appendix A (30, 60, 90, 120, and 150 min of ball-milling). It is observed in the DSC of the solid form of PGZ·HCl-RSV 1:1 at different ball-milling times that the ΔH_c_ (enthalpy of crystallization) values decrease as the time increases: (30 min: 39.57 J/g; 60 min: 24.26 J/g; 90 min: 16.29 J/g). Although a ΔH_c_ value (1.197 J/g) is still detected at 120 min, it must be considered that T_g_ is no longer observed, indicating that the coamorphous 1:1 form contains water molecules of hydration. However, it must be considered that as ΔH_c_ decreases as milling times increase, the endothermic enthalpy relaxation value increases: (30 min: 6.97 J/g; 60 min: 8.18 J/g; 90 min: 8.16 J/g). This implies that the longer the ball-milling time, the extent of semicrystalline PGZ·HCl in the coamorphous mixture decreases, but the relaxation of this drug is being promoted. This causes, after 120 min of milling, both components to lose their miscibility and gradually separate physically. As RSV separates, it allows the absorption of water molecules. It is well known that a coamorphous form can mainly establish hydrogen bonds between its participating components, which guarantees mixing between them, forming a single-phase amorphous form. Nevertheless, it seems that extending the milling times allows the miscibility between the constituents to be lost because these interactions are broken, permitting water molecules to establish hydrogen bonds with the components [20]. Again, as seen in the coamorphous 2:1 form, RSV cannot inhibit PGZ·HCl recrystallization.

The T_g_ value calculated (68.81 °C) with the Fox equation is above the experimental value (Table 3). For this reason, as mentioned with the coamorphous 2:1 form, it is assumed that there must be other factors apart from the interactions between the components not contemplated in the Fox equation, resulting in a higher-than-expected value.

In addition, the solid forms (1:2, 1:4, 1:6, 1:8, and 1:10) are characterized by DSC (Figure 6). Individual thermograms of DSC-TGA of all these solid forms are presented in Appendix A. The thermodynamic data are contained in Table 3.

All the molar ratios explored present the following T_g_ values: (1:2; 110.4 °C), (1:4; 112.7 °C), (1:6; 114.9 °C), (1:8; 115.2 °C), and (1:10; 117.3 °C). Additionally, all these solid forms present molecules of water of hydration according to the TGA. There is a large discrepancy between the calculated values and those determined experimentally (Table 3). It is difficult to attribute this difference to intermolecular interactions not contemplated in the theory of the Fox equation. It is proposed that in these outcomes, two solid forms coexist: the coamorphous form (either 1:1 or 2:1) and unreacted amorphous RSV. This is suggested for two reasons: (1) pure RSV is milling at different times, and within the first 30 min, a T_g_ event appears at 120.75 °C (Appendix A). As can be seen in Appendix A, the RSV that is used at the beginning is anhydrous and presents its T_g_ at 72.8 °C. Additionally, it presents a relaxation enthalpy of 128.9° (T_onset_), which is very close to the T_g_ observed when the RSV is ball-milled. It should also be noted that through ball-milling, the RSV hydrates, and that is why the change in T_g_ values occurs; (2) in Section 3.1.3, the SEM images show two different types of grain morphology, indicating that two solid forms coexist. Thus, the expected experimental T_g_ value for the coamorphous form in each of these samples cannot be observed, as it overlaps with the endotherms corresponding to the water of hydration molecules. Once again, there are discrepancies between the calculated T_g_ values and the experimental ones corresponding to the coamorphous form (Table 3). We attribute this to the fact that the coamorphous form is mixed with amorphous RSV.

Something to emphasize is that comparing the ΔH_m_ of the RSV pure (Table 1), it is much higher than any of these solid forms. In fact, it is observed that by increasing the %w RSV, there is a systematic decrease in the ΔH_m_ values (Table 3). It has been mentioned that larger enthalpies of mixing, either positive or negative, generally lead to significant non-idealities [20]. As the RSV proportion increases, the two components become more miscible given the values of ΔH_m_. 

With these results, when both drugs are in close stoichiometric ratios (2:1 or 1:1), from DSC analysis, the presence of T_g_ is observed, which indicates that under these conditions, both components have high miscibility, forming a single-phase amorphous form. In the case of the other stoichiometric ratios, the simultaneous coexistence of the coamorphous form and amorphous RSV is proposed.

Additionally, these solid forms are analyzed by FT-IR, and the bands −C=O_PGZ+_ and −C=O_RSV-_ are evaluated in Table 4 and Figure 7. The complete spectra of all these outcomes are shown in Appendix A.

Initially, observing the vibration -C=O_RSV-_, no shifts are perceived comparing pure RSV and RSV in the different solid forms. This indicates that the expected synthon −COO^−^···^+^H-N_pyr_ is not established in any outcome. In the specific case of the solid form 1:1, according to DSC, it is coamorphous. However, it is not a salt. This corroborates that in no case of the diffractograms is the presence of CaCl_2_ as a by-product observed. Likewise, as observed in the molar ratio of 2:1, the two components form a single-phase amorphous form (coamorphous), but it is not a salt. As the %w of RSV in the samples increases, the band corresponding to −C=O_RSV-_ increases in intensity. Shifts are observed in the vibration modes of −−C=O_PGZ+_ (a and b). However, they are not significant to consider that the coamorphous salt is formed. As %w of RSV increases and PGZ·HCl remains constant, the intensity of −C=O_PGZ+_ (b) decreases.

#### 3.1.2. Evaluation of the Formation of the PGZ·HCl-RSV Solid Forms (4:1, 6:1, 8:1, and 10:1)

Inspecting the diffractograms of these solid forms, something interesting is observed (Figure 8). The stoichiometric ratio of RSV remains constant, and PGZ·HCl is varied. It should be noted that the diffractogram of the solid form of PGZ·HCl -RSV (2:1) is added for comparison purposes.

In all cases, characteristic reflections of PGZ·HCl are seen. Although a large amorphous contribution is also seen in the outcomes, without a halo being observed, as the %w of PGZ·HCl increases, the intensity of these reflections increases. 

The DSC results of the stoichiometries (4:1, 6:1, 8:1, and 10:1) are shown in Figure 9. The individual DSC-TGAs of these proportions are shown in Appendix A. Initially, all DSC thermograms show two thermal events. The first is attributed to water molecules of hydration. This is corroborated by TGA. It should be emphasized that as the %w of PGZ·HCl in the samples increases, the ΔH of the hydration value of this thermal event decreases (4:1 (10.1 J/g); 6:1 (5.2 J/g); 8:1 (5.3 J/g); 10:1 (3.5 J/g)).

In the case of these molar ratio outcomes, it is possible to observe T_g_ experimental values that are very similar to those obtained in the coamorphous form (1:1 and 2:1) (Table 5 and Appendix A). However, these values are below the T_g_ values calculated with the Fox equation (Table 5). In this sense, it is thought that, as is observed in the exploration of the proportions where RSV is varied, and PGZ·HCl is left constant, two solid phases coexist in these samples, which causes these values to be above the experimental values. It is suggested that two solid forms simultaneously reside in these samples, the coamorphous form (either 1:1 or 2:1) and unreacted PGZ·HCl. This is proposed because (1) in the FT-IR results, as the %w of PGZ·HCl in the samples increases, the band corresponding to −C=O_PGZ+_ (b and b′) splits, indicating the presence of two species of H-PGZ^+^ (the one that participates in the coamorphous form and PGZ·HCl); (2) in the second endothermic event in the DCSs, it is observed that it contains two overlapping peaks that are due to the coamorphous form and unreacted PGZ·HCl. The observation of overlapping peaks is more evident in the PGZ·HCl-RSV 10:1 sample. Two T_m_ values are observed, but the first (159.6 °C) is like that observed in the coamorphous 1:1 form.

It is important to note that the experimental T_g_ values found for the coamorphous form in the solid forms (4:1, 6:1, 8:1, and 10:1**)** do not present the enthalpic relaxation endotherm observed in the coamorphous 2:1 and 1:1 forms (Appendix A). This indicates that the molecular mobility of amorphous drugs is reduced in the presence of unreacted PGZ·HCl.

The FT-IR spectra of these stoichiometry ratios are shown in Figure 10 and Table 6. The full spectra are shown in Appendix A. Initially, the vibration bands −C=O_PGZ+_ (a and b), although the changes in the Δν values are insignificant, indicate intermolecular interactions between the components. However, it is observed that as the %w of PGZ·HCl in the samples increases, the vibration -C=O_PGZ+_ (b) unfolds into two bands. This indicates that there are two types of H-PGZ^+^ in the solid phase, one that participates in the formation of the coamorphous 1:1 form, and the other as unreacted PGZ·HCl. This corroborates what is seen in DSC.

Regarding the band −C=O_RSV_^−^, changes in the Δν values are observed, but they do not convincingly indicate the formation of the synthon −COO^−^···^+^H-N_pyr_ to consider that coamorphous salt is formed. This supports that in XRPD, the presence of the expected CaCl_2_ by-product is not observed. Again, based on all the evidence, the coamorphous form must form in the solid phase (not in salt form) mixed with unreacted PGZ·HCl.

#### 3.1.3. SEM

The grain morphology of the solid forms (1:1, 1:6, 1:10, 6:1, and 10:1) is inspected and compared with the pure drugs (Table 7). SEM images are in Appendix A.

The grain morphology of the pure drugs is RSV (irregular) and PGZ·HCl (prismatic). In the case of the coamorphous 1:1 form, they do not present a defined morphology, although they resemble poorly defined prismatic forms. In the case that the %w of RSV is increased (1:6 and 1:10), two types of grains are observed: one irregular and one rod-shaped. This corroborates what has been seen in DSC that it is a mixture of solid phases, the coamorphous form and unreacted amorphous RSV. For the solid forms where the %w of PGZ·HCl is increased (6:1 and 10:1), the grain presents a poorly defined prismatic shape. It is not evident to observe two types of grains, to confirm what has been seen in DSC of having a mixture of solid phases.

#### 3.1.4. Determination of Dissolution Profiles and Solubility Studies

Dissolution studies are performed for the solid forms (1:1, 2:1, 1:4, 6:1, and 1:10), as well as the pure drugs PGZ·HCl and RSV. Initially, the IDR studies are carried out in the medium recommended for RSV by USP, which is citrate buffer solution at pH 6.6 [19]. Unfortunately, it is impossible to quantify RSV in this medium as PGZ·HCl precipitates immediately, disturbing the release of RSV into the solution. In this way, we carry out the studies in the medium recommended for PGZ·HCl, where both drugs can be simultaneously quantified (see Section 2.2.9). Binary coamorphous systems have been reported where the components are released synchronously into the dissolution medium [32]. This is attributed to the fact that the components establish short-range intermolecular interactions; mainly, centrosymmetric synthons (homo or hetero) are formed through hydrogen bonds. This phenomenon of simultaneous release of both components has been observed in the coamorphous mixtures of naproxen-cimetidine [28] and indomethacin-naproxen [33]. In the case of the first coamorphous blend, the homosynthon −COOH···HOOC− is established. In the case of the second, the heterosynthon imidazole ring···HOOC− is formed.

It has been proposed that the synchronized release is a consequence of the fact that both components present strong intermolecular interactions. For example, the addition of proline to the naproxen-tryptophan coamorphous binary system enhanced intermolecular interactions in the form of hydrogen bonding and increased the IDR of naproxen [34]. An improvement in the IDR profiles of indomethacin was also observed when it was mixed with amino acids to form the coamorphous system. These mixtures were revealed to show strong salt/partial salt interactions between the drug and amino acids, which helped to enhance the dissolving effect [26]. It is also mentioned that in these two coamorphous systems, the dissolution rate of the poorly soluble constituent depends on the solubility of the coformer. The coformer can then facilitate the improvement in the IDR profile of the test drug [32].

On the other hand, coamorphous systems are solid forms that present decreased crystal lattice energy values compared to the initial components [9]. This alters the lattice energy of the coamorphous form as the components are in an amorphous state within the solid phase. This modifies the dissolution and solubility properties of the components as they lack a crystal lattice arrangement. This favors the spontaneous release of the molecules into the medium. This scenario is known as the “spring effect” where large amounts of the drug are released into the solution. The duration of this “spring effect” will depend on the tendency of the drug being released to recrystallize. In other words, supersaturation may be limited if, in solution, the transformation (amorphous → crystalline) of the drug occurs rapidly, and, as mentioned above, the rate of dissolution depends additionally on the coformer, as this can delay and prevent the nucleation and crystal growth of the drug, slowing down the crystallization process [11]. If the coformer allows the recrystallization process to slow down, this situation is known as the “parachute effect” [7,35,36].

In this way, when evaluating the intrinsic dissolution rates (IDR) of the solid binary forms, they present the synchronized release of both components in the medium, even when strong intermolecular interactions between the components are not established, as has been seen in other systems. With respect to Figure 11, the IDR profiles of RSV release are presented in Table 8. Comparing the K_int_ values of the two coamorphous forms (2:1: 0.04191 ± 0.00901; 1:1: 0.08922 ± 0.00378), these are lower than the pure drug RSV (0.15475 ± 0.00429) (Table 8). It should be considered that this medium is not the one suggested for RSV (hydrochloric acid and potassium chloride buffer pH = 2.0). It has been mentioned that FDA-recommended dissolution conditions for RSV should be maintained at pH 6.6. In dissolution media with pH 1.2 and 4.5, the presence of RSV degradation products has been observed [37]. In our chromatographic determinations, more peaks than expected (unidentified analytes) are detected, which are attributed to these degradation products from RSV. It has already been described in detail by means of a study using an HPLC regarding the degradation products that RSV presents when found in acidic media [38].

Based on this, it is suggested that in this acid medium, the RSV anion is protonated (calcium salt → acid form), which changes its solubility and does not allow the supersaturation effect once the coamorphous form is introduced into the medium, and in parallel, as this medium is indicated for PGZ·HCl, this drug must be dissolved. Apparently, if PGZ·HCl remains dissolved, it does not disturb the RSV release process. Given this situation, if the solubility of RSV changes due to its protonation, the supersaturation effect is limited, not observing a sudden enrichment of the drug concentration in the medium. For this reason, the K_int_ of the coamorphous form is not higher than that of the pure RSV. The coamorphous 1:1 form presents a higher K_int_ value than the coamorphous 2:1 form, because the former presents a higher %w of RSV (1:1: 55.58%) and (2:1: 38.76%).

In the case of the solid forms of PGZ·HCl-RSV (1:4, 6:1, and 1:10), the binary mixture 6:1 has the lowest %w of RSV (17.26%) and is the one that presents the lower K_int_ (0.02953 ± 0.00329). For the other solid forms, the %w of RSV varies as follows: PGZ·HCl-RSV (1:4 (83.60%) and 1:10 (92.61%)). It is unclear why the one containing the lower %w of RSV has a higher K_int_ value (0.12320 ± 0.00153 and 0.11724 ± 0.01791).

In the case of the IDRs of PGZ·HCl release, the following is observed (Figure 12). It should be emphasized that PGZ·HCl is a BCS II drug [39]. According to the K_int_ values (Table 8), the coamorphous 2:1 system (0.07324 ± 0.00691) presents a value 1.03 times higher with respect to pure PGZ·HCl (0.07076 ± 0.00317). This indicates that in this coamorphous system (2:1), the “spring effect” is observed, but this is limited because, as mentioned, if there is the simultaneous release of both drugs into the medium, if RSV is protonated, it consumes protons from the medium that causes the pH to vary. Therefore, this pH variation limits the supersaturation effect. It has been reported that the solubility of PGZ·HCl depends on the type of buffer and, thus, the pH value [40]. Comparing both coamorphous forms (2:1 (61.24%) and 1:1 (44.42%)), considering the %w of PGZ·HCl, both present very similar K_int_ values (0.06970 ± 0.00269). In these cases, it is not observed that the dissolved amount of PGZ·HCl depends on the %w. That is why it is suggested that the “spring effect” exists but that it is affected by the change in pH.

In the solid forms that are not coamorphous (1:4, 6:1, and 1:10), they present much lower K_int_ values (0.12320 ± 0.00153; 0.02953 ± 0.00329; 0.11724 ± 0.01791). Of these, the binary mixture (6:1: 82.74%) is the one that presents the highest K_int_. Of the other two (1:4: 16.40%; 1:10: 7.39%), we do not find an explanation for why they present such similar K_int_ values. It should be noted that with the 6:1 solid form, even having the %w of PGZ·HCl higher than those of coamorphous forms (2:1 or 1:1), the latter has a higher release behavior due to the higher K_int_ values. This corroborates what was previously described: these coamorphous forms present a “spring effect” limited by the pH variation due to the consumption of protons from the medium by the RSV degradation products.

Attempts are made to determine the solubility of the pure drugs and the various binary solid forms in the medium recommended for PGZ·HCl; however, no consistent results are obtained. We attribute this to the fact that, in addition to the fact that RSV presents degradation compounds in an acidic medium, it has been reported that PGZ·HCl also decomposes when it remains for long periods of time [41].

## 4. Conclusions

Ball-milling turned out to be a versatile synthetic tool in preparing the solid form of PGZ·HCl-RSV in a 2:1 molar ratio using NG or LAG with different solvents. These mechanochemical reactions allowed for obtaining: NG (coamorphous), water (physical mixture), and EtOH and hexane (not defined). It should be emphasized that in the case of the solid form, AcOEt did not completely favor the formation of the coamorphous form as was achieved in NG. It was confirmed by FT-IR studies that this coamorphous 2:1 form was not a salt.

Other stoichiometries were explored, either keeping PGZ·HCl constant and varying RSV or vice versa; only in the case of a molar ratio 1:1 could another coamorphous form be obtained. It was found that the other solid forms explored were mixtures containing, simultaneously in the solid phase, the coamorphous form (either 2:1 or 1:1) together with an excess drug. When RSV was in excess, it was in an amorphous form. In the case of PGZ·HCl, it was found in a semi-crystalline form.

IDRs of the solid forms of PGZ·HCl -RSV (1:4, 6:1, and 1:10) as well as the coamorphous forms (2:1 and 1:1) were evaluated, and these were compared with the pure drugs RSV and PGZ·HCl. Unfortunately, the IDRs could not be determined in the medium recommended by USP for RSV, due to PGZ·HCl solubility problems, which disturbed the statin release process. Therefore, the dissolution medium recommended by the USP for PGZ·HCl was used as an acidic medium. It should be noted that in the dissolution profile studies, it was found that both drugs were released into the medium in a synchronized manner. No improvement was observed in the dissolution profiles of any of the solid forms tested compared to the IDR of pure RSV. This was because it has been pointed out that RSV in acidic media presents the formation of degradation products. However, as the suitable medium for PGZ·HCl, the coamorphous 2:1 form, it presented an improvement of 1.03 times with respect to the pure drug. Based on the results, we propose that the supersaturation effect typically observed in the coamorphous form was limited because RSV consumed protons from the medium, which caused the pH of the medium to vary, changing the solubility of PGZ·HCl. It is well known that the solubility of this drug depends on the buffer used and the pH of the medium.

Based on this, PGZ·HCl is not recommended as a coformer to form coamorphous systems, mainly because of its reluctance to remain amorphous. 

## Data Availability

No new data were created or analyzed in this study. Data sharing is **n**ot applicable to this article.

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
