# Peer review of "Ball-Milling Preparation of the Drug–Drug Solid Form of Pioglitazone-Rosuvastatin at Different Molar Ratios: Characterization and Intrinsic Dissolution Rates Evaluation"

_pharmaceutics, 2023, doi:10.3390/pharmaceutics15020630_

Round 1

Reviewer 1 Report

The authors designed a coamourphous solid of RSV and PGZ by ball-milling with or without solvents. The solid-state properties of the produced solid were characterized by FTIR, PXRD and DSC. The dissolution of produced solid was also compared and discussed. My suggestions are listed as follows.

1.      In Abstract, only FTIR and DSC results were mentioned. Since PXRD results were also crucial to identify the amorphous state, and were investigated in this study. The discussion of PXRD results should be included.

2.      In section 2.1, the purity of two drugs and all solvents used in this study should be included.

3.      In section 2.2, hexane, ethyl acetate and ethanol were used in milling process, especially for hexane, it may bring concern regarding the residual solvent contamination. Please give comments on the residual solvent issue.

4.       In section 3.1 and Figure 1, please give more discussion regarding the results from CaCl2. The general readers for Pharmaceuticals may confuse why PXRD pattern of CaCl2 is crucial in this section.

5.      In addition to the crystal structure properties, please also provide SEM images and particle size results to show the comparison of crystal habit and particle size characteristics before and after the coamorphous process.

6.      In dissolution study, the authors mentioned an improvement of 1.03 times (about 3%) with respect to the pure drug. However, from Table 8, the deviation of dissolution data is about 5-10%. That means the dissolution improvement is insignificant compared with the experimental uncertainty. In addition, ball milling is also a particle size reduction process to improve dissolution behavior, the author combines two strategies (coamourphous process and micronization) for dissolution improvement but the beneficial effect of these two process did not found. The authors should reconsider the motivation and significance of this study in addition to dissolution improvement.

Author Response

We appreciate the reviewer's positive comments and his/her close reading of the manuscript. According to the reviewer’s comments, the manuscript has been thoroughly revised and corrected where necessary.

Please find below the specific answers to her/his observations.

Reviewer 2 Report

The work by Tecocoatzi et al. on the use of ball milling to prepare amorphous systems is generally well presented with the exception of the dissolution tests, but some clarifications are required, as follows.

The methodology proposed is fine although other techniques (e.g. NMR either liquid or solid states) would provide further useful information on the elucidation of the formation of the amorphous systems.

In fact, the absence of a satisfactory elucidation of the mechanism of interaction between the two drugs is the weakest part of the work and from this, other questions arise, namely:

a)      What is the rationale to suggest a large number of molar ratios between the 2 drugs,

b)      Apparently a 1:1 proportion should have provided the best outcome;

c)       Did the authors consider using the free forms of the drugs? The use of salts was likely to prevent the formation of a new salt, particularly when the raw material salts (HCl and Ca) are very stable. Any suggestion why CaCl2 failed to be detected and quantified?

d)      Authors claim the usefulness of amorphization in increasing the solubility of the drugs. However, the outcome was not significant. Please provide an explanation;

e)      It is not clear why the authors decided to use a low pH in the dissolution tests. Furthermore, have the authors considered non-sink conditions?

Author Response

(The authors gave the same response as above.)

Round 2

Reviewer 1 Report

I think the authors have responded my comments appropriately.